

# Evaluation of different culture media to support in vitro growth and biofilm formation of bacterial vaginosis-associated anaerobes

Aliona S. Rosca, Joana Castro and Nuno Cerca

Laboratory of Research in Biofilms Rosário Oliveira—Centre of Biological Engineering,
University of Minho, Braga, Portugal

## ABSTRACT

**Background:** Bacterial vaginosis (BV) is one of the most common vaginal infections worldwide. It is associated with the presence of a dense polymicrobial biofilm on the vaginal epithelium, formed mainly by *Gardnerella* species. The biofilm also contains other anaerobic species, but little is known about their role in BV development.
**Aim:** To evaluate the influence of different culture media on the planktonic and biofilm growth of six cultivable anaerobes frequently associated with BV, namely *Gardnerella* sp., *Atopobium vaginae*, *Lactobacillus iners*, *Mobiluncus curtisii*, *Peptostreptococcus anaerobius* and *Prevotella bivia*.
**Methods:** A total of nine different culture media compositions, including commercially available and chemically defined media simulating genital tract secretions, were tested in this study. Planktonic cultures and biofilms were grown under anaerobic conditions (10% carbon dioxide, 10% helium and 80% nitrogen). Planktonic growth was assessed by optical density measurements, and biofilm formation was quantified by crystal violet staining.
**Results:** Significant planktonic growth was observed for *Gardnerella* sp., *A. vaginae* and *L. iners* in New York City III broth, with or without ascorbic acid supplementation. Biofilm quantification showed high in vitro biofilm growth for *Gardnerella* sp., *P. anaerobius* and *P. bivia* in almost all culture media excluding Brucella broth. Contrary, only New York City III broth was able to promote biofilm formation for *A. vaginae*, *L. iners* and *M. curtisii*.
**Conclusions:** Our data demonstrate that New York City III broth relative to the other tested media is the most conducive for future studies addressing polymicrobial biofilms development as this culture medium allowed the formation of significant levels of single-species biofilms.

Corresponding author
Nuno Cerca,
nunocerca@ceb.uminho.pt

## INTRODUCTION

Bacterial vaginosis (BV) is the worldwide leading bacterial vaginal infection commonly recognized in women of all ethnicities between menarche and menopause (*Beamer et al., 2017*; *Javed, Parvaiz & Manzoor, 2019*). BV is often characterized by a decrease of

beneficial vaginal bacteria, mainly hydrogen peroxide and lactic acid-producing *Lactobacillus* species, and by an increase of anaerobic pathogens (*Marrazzo, 2011*; *Schwebke, 2000*). The most prominent of these is *Gardnerella vaginalis*, a facultative anaerobe usually found embedded in a polymicrobial biofilm (*Swidsinski et al., 2014*). It is important to mention that an emended description of *G. vaginalis* was recently proposed with the delineation of 13 genomic species within the genus *Gardnerella* (*Vaneechoutte et al., 2019*). Of these 13 species, three were officially described (*G. leopoldii*, *G. piotii*, and *G. swidsinskii*) and *G. vaginalis* was amended. Following this renewed taxonomy of the genus *Gardnerella*, in this article, the term *Gardnerella* spp. will be used to address previous publications, since we cannot rule out the fact that other *Gardnerella* species were involved.

According to the current hypothesis for BV pathogenesis, *Gardnerella* spp. initiate the formation of the biofilm on vaginal epithelial cells and become a scaffolding to which other BV-associated species thereafter can attach (*Muzny et al., 2019*). One of the species that is often found associated with *Gardnerella* spp. biofilms is *Atopobium vaginae* (*Swidsinski et al., 2005*). Under specific in vitro conditions, *G. vaginalis* enhances culturability of *A. vaginae* (*Castro et al., 2020*) and it has been suggested that the in vivo detection of both bacteria is a strong indicator of BV development (*Bradshaw et al., 2006*; *Hardy et al., 2016*, *2015*; *Muzny et al., 2019*). However, the BV biofilm is often populated by many other facultative or strict anaerobes, but very little is known about their role in BV development. As such, more studies are needed to address the interactions between these species. One issue facing researchers that work with BV-associated species is that most species are uncultivable or fastidious (*Diop et al., 2019*; *Fredricks et al., 2007*; *Fredricks, Fiedler & Marrazzo, 2005*; *Srinivasan et al., 2016*). Furthermore, in vitro biofilm formation requirements are often different from planktonic growth (*Alves et al., 2014*). Considering the increased focus on biofilm-associated infections and the demand for finding novel treatment approaches (*Falconi-McCahill, 2019*), the current study was undertaken aiming to evaluate the effects of nine different culture media on the planktonic growth and biofilm formation of cultivable anaerobes frequently found in BV, namely *Gardnerella* sp., *A. vaginae*, *Mobiluncus curtisii*, *Peptostreptococcus anaerobius* and *Prevotella bivia* (*Diop et al., 2019*; *Onderdonk, Delaney & Fichorova, 2016*). Importantly, *Lactobacillus iners* was also included in this study as this species plays a controversial role in the vaginal microenvironment, being detected in the vaginal microbiota of both healthy (*Anukam et al., 2006*; *Ravel et al., 2011*) and women with BV (*Rampersaud et al., 2011*). Furthermore, *L. iners* has been often identified in the intermediate vaginal microbiota (i.e., between normal and BV microbiota) (*Jakobsson & Forsum, 2007*; *Petrova et al., 2015*) and also dominates the microbiota after treatment of BV (*Ferris et al., 2007*). Also, this microorganism has complex nutritional requirements, being thus, not easy to work with in vitro (*Vaneechoutte, 2017*).

As such, the main goal of this study was to explore growth conditions optimal for using in future in vitro multi-species biofilm model, with the aim to better analyze BV multi-species interactions and their impact on BV development.

**Table 1** Cultivable bacterial species used for planktonic and biofilm growth assays.

| Species | Strain | Origin | Association with BV[1] |
|---|---|---|---|
| *Gardnerella* sp. | UM241[2] | Isolated from women diagnosed with BV | Often described |
| *Atopobium vaginae* | ATCC BAA-55[T] | Isolated from vaginal microbiota of a healthy woman (*Jovita et al., 1999*) | Often described |
| *Lactobacillus iners* | CCUG 28746[T] | Isolated from human urine (*Falsen et al., 1999*) | Commonly described |
| *Mobiluncus curtisii* | ATCC 35241[T] | Isolated from women with BV (*Spiegel & Roberts, 1984*) | Commonly described |
| *Peptostreptococcus anaerobius* | ATCC 27337[T] | Isolated from female genital tract (*Ng et al., 1994*) | Commonly described |
| *Prevotella bivia* | ATCC 29303[T] | Isolated from endometrium (*Holdeman & Johnson, 1977*) | Commonly described |

Notes:
[1] To determine how often the selected cultivable species have been reported in BV, a query in PubMed was performed by using a specific combination of keywords as "bacterial vaginosis" and "name of each species" (e.g., "*Gardnerella*"and "bacterial vaginosis"). We designated as "often described" those species referred in more than 50 articles in the last 25 years, while "commonly described" had at least 10 articles in the same period. Of note, the selected bacterial species used herein have been pointed out in several studies (*Diop et al., 2019*; *Onderdonk, Delaney & Fichorova, 2016*; *Ravel et al., 2011*) as potential microbial pathogens involved in BV development.
[2] The partial 16S ribosomal RNA gene sequences of *Gardnerella* sp. is downloadable from NCBI. UM: University of Minho, Portugal. In addition, the strain was phenotypically and genotypically characterized by *Castro et al. (2020, 2015)*, *Castro, Jefferson & Cerca (2018)*. It is of note that *Gardnerella* sp. UM241 did not match with any of the *Gardnerella* species described by Vaneechoutte and colleagues (*Vaneechoutte et al., 2019*) (i.e., *G. vaginalis*, *G. piotii*, *G. leopoldii* and *G. swidsinskii*), belonging to a yet unidentified *Gardnerella* species (*Castro et al., 2020*).

## MATERIALS AND METHODS

### Bacterial species and growth conditions

Six cultivable bacterial species associated with BV were used in the current study, namely *Gardnerella* sp., *A. vaginae*, *L. iners*, *M. curtisii*, *P. anaerobius* and *P. bivia* (Table 1). These species were preserved frozen in Brain Heart Infusion broth (BHI) (Liofilchem, Italy) with 23% (v/v) glycerol (Panreac, Spain) at −80 °C. Each species was inoculated from the −80 °C bacterial stock on plates containing Columbia Blood Agar medium (CBA) (Oxoid, Basingstoke, UK) supplemented with 5% (v/v) defibrinated horse blood (Oxoid, Basingstoke, UK) and incubated at 37 °C under anaerobic conditions [controlled atmosphere composed of 10% carbon dioxide ($CO_2$), 10% helium and 80% nitrogen generated by a cylinder (Air Liquid, Algés, Portugal) coupled to an anaerobic incubator (Plas-Labs, Lansing, MI, USA)] for 2–4 days. For planktonic and biofilm assays, Brain heart infusion broth supplemented with yeast extract, starch and gelatin (sBHI), Brucella broth supplemented with hemin and vitamin $K_1$ solutions (BHV), New York City III broth supplemented with 10% (v/v) inactivated horse serum (NYC), Schaedler broth (SB), and a medium simulating genital tract secretions (mGTS) were used as culture media with the mentioned composition, but also supplemented with 0.1% (w/v) L-ascorbic acid (Sigma–Aldrich, Gillingham, UK), excepting mGTS which already contains L-ascorbic acid. The addition of L-ascorbic acid to the culture media was designated with the abbreviation "Aa", added at the end of each medium's name mentioned above (e.g., sBHI supplemented with L-ascorbic acid became sBHI.Aa). The detailed information about each tested medium is presented in Table 2. In order to prepare hemin solution, 0.5 g of hemin was dissolved in 10 mL of 1 N NaOH and afterwards distilled water was added to reach the volume of 100 mL. Further, vitamin $K_1$ solution was prepared by adding 0.025 mL of vitamin $K_1$ stock solution to 4.975 mL of 95% ethanol. The prepared solutions of hemin and vitamin $K_1$ were used with a concentration of 0.0005% (w/v) and 0.0001% (w/v), respectively. In mGTS, Part III of this medium is a vitamin mixture,

**Table 2 Culture media used for the growth of BV-associated anaerobes.**

| Culture medium | Composition | Supplementation | Abbreviation |
|---|---|---|---|
| Brain heart infusion broth (Liofilchem, Italy) | As described by the manufacturer | 2% (w/w) Gelatine (Liofilchem, Italy); 0.1% (w/w) Starch (Panreac, Spain); 0.5% (w/w) Yeast extract (Liofilchem, Italy) | sBHI/sBHI.Aa[1] |
| Brucella broth (Liofilchem, Italy) | As described by the manufacturer | 0.0005% (w/v) Hemin (Sigma, China); 0.0001% (w/v) Vitamin $K_1$ (Sigma, China) | BHV/BHV.Aa[1] |
| New York City III broth | 1.5% (w/v) Bacto proteose peptone no. 3 (BD, France); 0.5% (w/v) Glucose (Fisher Scientific, UK); 0.24% (w/v) HEPES (VWR, USA); 0.5% (w/v) NaCl (VWR, USA); 0.38% (w/v) Yeast extract (Liofilchem, Italy) | 10% (v/v) Inactivated horse serum (Biowest, France) | NYC/NYC.Aa[1] |
| Schaedler broth (Liofilchem, Italy) | As described by the manufacturer | – | SB/SB.Aa[1] |
| Chemically defined medium simulating genital tract secretions (*Stingley et al., 2014*) | Part I: 0.35% NaCl; 0.15% KCl; 0.174% $K_2HPO_4$; 0.136% $KH_2PO_4$; 1.08% glucose; 0.05% cysteine HCl. Part II: 0.1% glycogen; 0.03% mucin; 0.02% tween 20; 0.05% urea; 0.0005% hemin; 0.0001% vitamin $K_1$; 0.2% bovine serum albumin; 0.03% $MgSO_4$; 0.004% $NaHCO_3$; 0.1% sodium acetate; 0.005% $MnCl_2$. Part III: 0.0005% biotin; 5.0% *myo*-inositol; 0.05% niacinamide; 0.05% pyridoxine HCl; 0.05% thiamin HCl; 0.05% D-calcium pantothenate; 0.05% folic acid; 0.001% *p*-aminobenzoic acid; 0.05% choline chloride; 0.01% riboflavin; 0.1% L-ascorbic acid; 0.0005% vitamin A (retinol); 0.0005% vitamin D (cholecalciferol); 0.001% vitamin $B_{12}$. Part IV (amino acids): 0.032% alanine; 0.008% arginine; 0.076% aspartic acid; 0.036% glutamic acid; 0.04% glutamine; 0.02% glycine; 0.016% histidine; 0.012% isoleucine; 0.02% leucine; 0.02% lysine; 0.004% methionine; 0.004% phenylalanine; 0.028% proline; 0.012% serine; 0.012% threonine; 0.004% tryptophan; 0.02% tyrosine; 0.068% valine. Part V (UPI): 0.05% uracil; 0.01% sodium pyruvate; 0.02% inosine. | – | mGTS[2] |

Notes:
[1] The supplementation of the culture media with 0.1% (w/v) L-ascorbic acid was designated with the abbreviation "Aa", added at the end of each medium's name as follows, for example, sBHI supplemented with L-ascorbic acid became sBHI.Aa. The other three culture media were abbreviated by following the same rule.
[2] The mGTS medium has already 0.1% (w/v) L-ascorbic acid in its composition.

Sigma K3129, from Sigma–Aldrich (UK) with the stock solution of 100× that was used at a concentration of 0.5% (v/v).

## Planktonic growth assessment

For the evaluation of planktonic growth, the inoculums were prepared by transferring fresh bacterial colonies from CBA plates to 8 mL of each culture medium described above. The obtained bacterial suspensions were adjusted by optical density (OD) at 620 nm to 0.10 ± 0.05 (EZ Read 800 Plus; Biochrom, Cambridge, UK) and equally distributed in two
sterile 15 mL falcon tubes (Orange Scientific, Braine-l'Alleud, Belgium) which were further incubated at 37 °C under anaerobic conditions for 48 h, as described above. Afterwards, planktonic growth was assessed by $OD_{620nm}$. Growth was normalized as a fold difference between the final $OD_{620nm}$ and the starting OD (at time 0 h). The assays were repeated at least three times on separate days, with four technical replicates considered each time.

## Biofilm formation and biomass quantification

Single-species biofilms of each tested species were initiated by inoculating bacterial suspensions of 48 h cultures adjusted to an $OD_{620nm}$ of $0.10 \pm 0.05$ in sterile 96-well tissue culture plates (Orange Scientific, Braine-l'Alleud, Belgium) and incubated for 72 h, at 37 °C under anaerobic conditions. To quantify the biofilm biomass, we used the crystal violet (CV) method, which is the most frequently employed approach for this purpose (*Azeredo et al., 2017*; *Peeters, Nelis & Coenye, 2008*). In brief, following 72 h of incubation, the biofilms were washed once with 0.9% (w/v) sodium chloride and allowed to air dry. After, the biofilms were fixed with 100% (v/v) methanol (Thermo Fisher Scientific, Waltham, MA, USA) for 20 min, and then stained with CV solution 1% (v/v) (Merck, Darmstadt, Germany) for 20 min. Subsequently, each well was washed twice with 1% (v/v) phosphate-buffered saline, and the bound CV was released with 33% (v/v) acetic acid (Thermo Fisher Scientific, Lenexa, KS, USA). To assess the biomass, the OD of the resulting solution was measured at 595 nm. Biofilm experiments were repeated at least three times with eight technical replicates.

## Statistical analysis

The data were analyzed using the statistical package GraphPad Prism version 6 (La Jolla, CA, USA) by one-way ANOVA (Dunnett's and Tukey's multiple comparison tests) and two-way ANOVA (Sidak's multiple comparisons test). Values with a $p < 0.05$ and $p < 0.01$ were considered statistically significant.

# RESULTS

## Planktonic growth assays

As shown in Fig. 1, BV-associated anaerobes had variable ability to grow planktonically in the tested culture media. Accordingly, *P. anaerobius* and *P. bivia* had higher metabolic flexibility and were able to grow in most of the tested media, while *M. curtisii* had more restrictive growth requirements and presented low levels of growth in all of them. Interestingly, NYC broth showed high levels of planktonic growth for the tested species, being overpassed only by NYC.Aa for *L. iners*, SB and SB.Aa for *P. anaerobius*, and by sBHI.Aa for *P. bivia*. The mGTS supported very low levels of bacterial growth, with only *Gardnerella* sp., *P. anaerobius* and *P. bivia* showing moderate levels of growth.

Since it was previously shown that L-ascorbic acid could enhance the growth of several anaerobic bacteria, including *A. vaginae*, *Finegoldia magna*, *Fusobacterium necrophorum*, *Prevotella nigrescens*, *Ruminococcus gnavus* and *Solobacterium moorei* (*La Scola et al., 2014*), we repeated the experiments with media supplemented with 0.1% (w/v) L-ascorbic

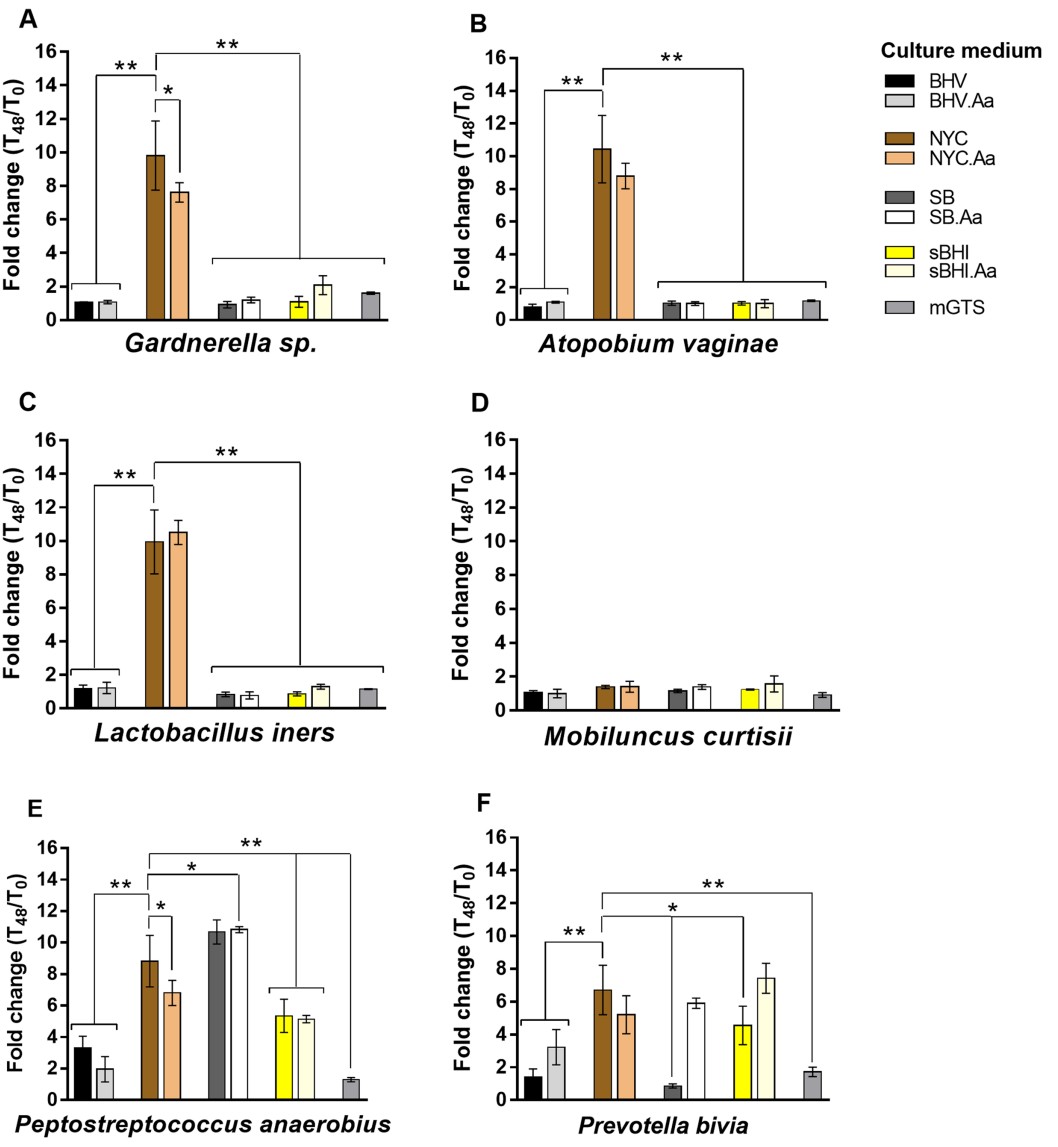

**Figure 1** **Fold change in planktonic growth of BV-associated bacteria in the nine different culture media relative to OD$_{620nm}$ values measured at T$_0$ h.** (A) Experiments conducted with *Gardnerella* sp. (B) Experiments conducted with *A. vaginae*. (C) Experiments conducted with *L. iners*. (D) Experiments conducted with *M. curtisii*. (E) Experiments conducted with *P. anaerobius*. (F) Experiments conducted with *P. bivia*. Results represent the average ± the standard deviation of at least three independent experiments. Statistical analysis was performed using one-way ANOVA and Dunnett's multiple comparisons test. Significant differences between NYC medium (our culture medium of choice) and other culture media are depicted with *$p < 0.05$ and **$p < 0.01$. Other statistical analysis is shown in Table S1.

acid. However, contrary to what was described before, the addition of 0.1% (w/v) L-ascorbic acid had a very variable effect on bacterial growth, with only 33.3% ($n = 8$; cut-off ≥ 1.25-fold change) of the total combinations tested yielding a significant increase in bacterial growth, while in 4.17% ($n = 1$; cut-off < 0.75-fold change) an inhibition of the growth was observed. In most of the tested combinations ($n = 15$; 0.75 ≤ fold change < 1.25), no effect was observed (Table S1). The most notable case was observed

for *P. bivia* growth in SB, in which L-ascorbic acid increased almost seven-fold the growth rate.

## Biofilm assays

Not surprisingly, we observed that similar to planktonic growth, biofilm formation was also strongly affected by the culture media composition, as depicted in Fig. 2. Importantly, there was not a direct relationship between higher planktonic growth and higher biofilm formation, which further confirms that the requirements for biofilm formation are distinct than the requirements for planktonic growth, as showed before for many other bacterial species (*Alves et al., 2014*; *Heffernan, Murphy & Casey, 2009*; *Ripolles-Avila et al., 2018*; *Wijesinghe et al., 2019*). Further differences between biofilm formation and planktonic growth were observed when adding L-ascorbic acid to the growth media, with 20.8% ($n = 5$) of the tested combinations species/growth medium resulting in a statistically significant decrease in the biofilm-forming capacity ($p < 0.05$) and 37.5% ($n = 9$) of the situations also presenting a visible biofilm reduction, however not statistically significant as can be seen in Table S2. Moreover, the addition of L-ascorbic acid to the culture media did not promote significantly increased biofilm formation in any of the combinations tested (Fig. 2).

## DISCUSSION

Despite the fact that BV is an increasingly important health problem, there is a lack of studies addressing multi-species interactions that might occur during BV and their role in the development of this infection. Most attempts to understand the microbiology behind BV have been focused mainly on *Gardnerella* spp., perhaps because this species has long been associated with BV development (*Gardner & Dukes, 1955*; *Swidsinski et al., 2005*) and it has been now hypothesized that this microorganism is the initial colonizer of the vaginal epithelium, being able to establish an early biofilm structure to which other BV-associated species can attach (*Muzny et al., 2019*). However, the role of these species in the development and progress of BV is still poorly understood and therefore, more studies are needed to unravel this matter.

We showed before that BV-associated species had different abilities to grow as biofilms, and this was strongly dependent on the growth media (*Alves et al., 2014*). As such, the first step in facilitating BV multi-species biofilm studies is to determine optimal culture medium conditions suitable for multiple BV-associates species, considering to further investigate the interactions that might exist between them in BV multi-species biofilms and their implications in BV process.

Although sBHI has been widely used as a medium that supports *Gardnerella* spp. growth (*Algburi, Volski & Chikindas, 2015*; *Gottschick et al., 2016*; *Harwich et al., 2010*; *Machado, Palmeira-de-Oliveira & Cerca, 2015*; *Patterson et al., 2010*; *Turovskiy et al., 2012*; *Weeks et al., 2019*), it did not facilitate the planktonic growth or biofilm formation for some of the tested species, including *A. vaginae*, *L. iners* and *M. curtisii*. The same was observed for these three species in SB medium, even though the manufacturer describes it as a medium suitable for the cultivation of anaerobic microorganisms, providing them

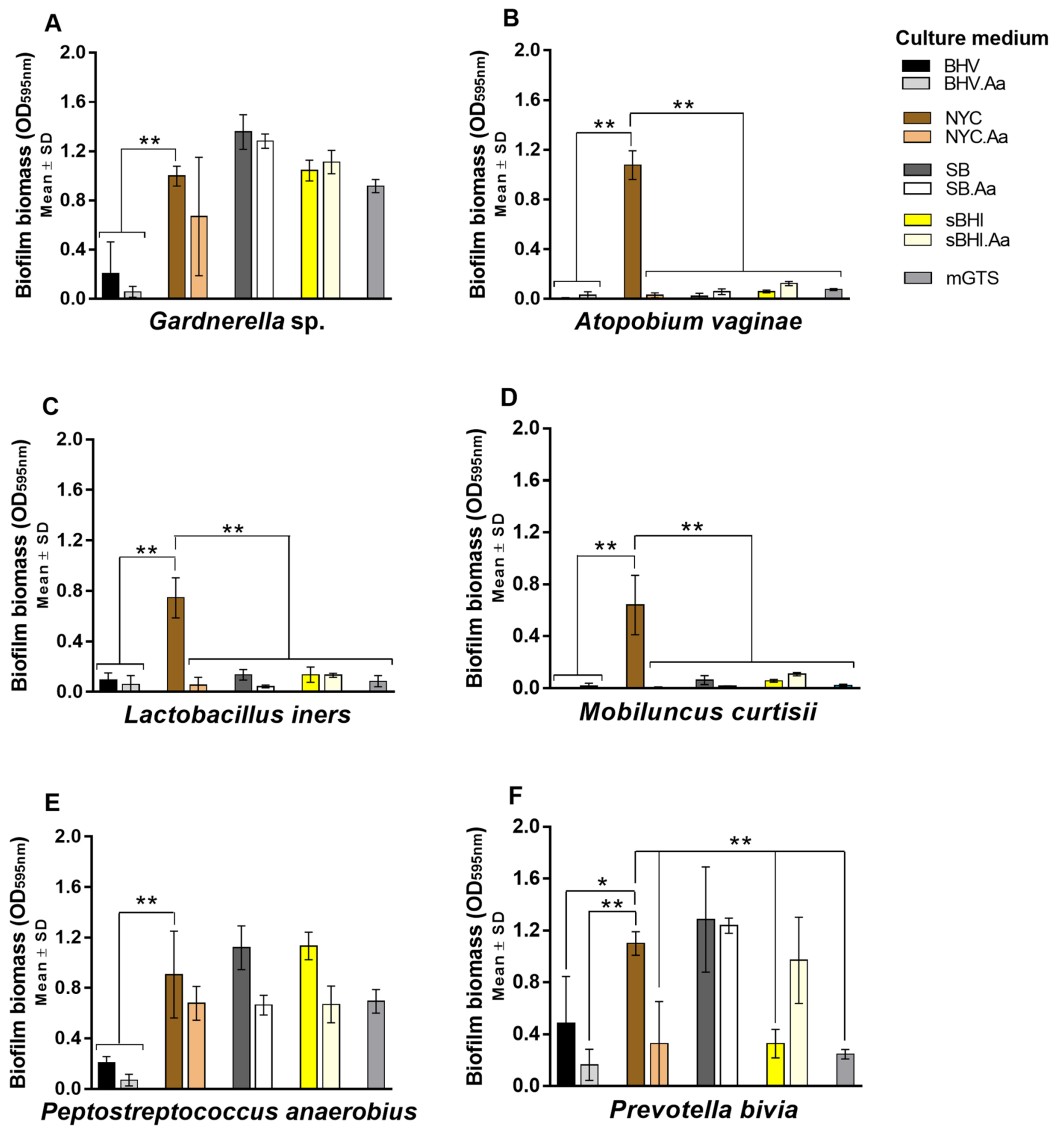

**Figure 2 Biofilm formation of BV-associated bacteria in the nine different culture media over a 72 h period.** Biofilm biomass was quantified using the crystal violet staining assay. (A) The total biofilm biomass formed by *Gardnerella* sp. in the tested culture media. (B) The total biofilm biomass formed by *A. vaginae.* (C) The total biofilm biomass formed by *L. iners.* (D) The total biofilm biomass formed by *M. curtisii.* (E) The total biofilm biomass formed by *P. anaerobius.* (F) The total biofilm biomass formed by *P. bivia.* Results are expressed as the average ± standard deviation of at least three independent experiments performed with eight technical replicates. Statistical analysis was performed using one-way ANOVA and Dunnett's multiple comparisons test. Significant differences between biofilm biomass formed in NYC medium (our culture medium of choice) and other culture media are represented with $*p < 0.05$ and $**p < 0.01$. Other statistical analysis is shown in Table S2.

an important amount of amino acids, nitrogen, vitamins as well as the energy necessary for growth. In an early study, after evaluating nine broth media in varied $CO_2$ atmospheres for their ability to support the growth of anaerobic bacteria including *Bacteroides fragilis* subspecies *fragilis*, *Peptostreptococcus* CDC group 2, *Eubacterium alactolyticum* and *Clostridium perfringens*, *Stalons, Thornsberry & Dowell (1974)* found that SB in an

atmosphere of 5% $CO_2$, 10% hydrogen and 85% nitrogen exhibited the fastest and highest growth response. However, in our in vitro conditions, we obtained high levels of planktonic growth only for *P. anaerobius*, probably because this medium is not appropriate for the growth of all species of anaerobic microorganisms. Still, SB was a good medium to support in vitro biofilm formation with high levels of the biomass for *Gardnerella* sp., *P. anaerobius* and *P. bivia*. Interestingly, *Gardnerella* sp. and *P. bivia* showed in SB the lowest levels of planktonic growth, but the highest biofilm formation ability. As mentioned, SB is a complex medium and perhaps the presence of certain growth factors determined these two anaerobes to turn on the expression of biofilm-related genes. Another of the tested media, BHV, also described by the manufacturer as suitable for the cultivation of anaerobes, was not appropriate for the growth and biofilm formation of the tested species, with the exception of planktonic growth by *P. anaerobius* (Fig. 1).

Interestingly, NYC facilitated the planktonic growth of all tested species, despite *M. curtisii* presented a very slow growth rate. Nevertheless, even *M. curtisii* was able to form high biofilm biomass in this medium. In fact, together with *A. vaginae* and *L. iners*, significant biofilm formation was only detected in NYC (Fig. 2). A particularity of NYC medium, compared to the other tested media, is the presence of proteose peptone no. 3, which has been described by the manufacturer as offering high nutritional benefits to fastidious anaerobic species by providing the necessary amount of nitrogen, carbon, amino acids and essential growth factors. To assess if, in fact, the enhancement of biofilm formation in NYC was mainly due to the presence of proteose peptone no. 3, we carried out an experiment by evaluating the biofilm-formation ability of the six tested bacterial species in the original recipe of NYC *versus* an altered version of NYC [with regular peptone from meat (Acros Organics, UK) replacing the proteose peptone no. 3]. Interestingly, while we did find that proteose peptone no. 3 was essential to the biofilm formation by *M. curtisii*, no significant differences were found for the other species (Fig. S1), which suggests that NYC's ability to enhance biofilm formation is not only related to the presence of proteose peptone no. 3.

Besides the commercially available media, we also tested a chemically defined medium that simulates the genital tract fluid, mGTS (*Stingley et al., 2014*). Since mGTS is a minimal medium without rich nutrient sources, it was not surprising that the growth of the tested BV-associated species was negligent or very slow in this medium. Nevertheless, biofilm formation by *Gardnerella* sp. and *P. anaerobius* was significant under mGTS, further being confirmed that biofilm formation requires specific conditions, different from planktonic growth.

We also tested another variable in our growth conditions optimization. The addition of L-ascorbic acid had the advantage of reducing the oxidation potential of the growth media by removing the oxygen (*La Scola et al., 2014*). However, the effect of adding L-ascorbic acid was very variable, depending not only on the bacterial species but also on the respective growth media. Nevertheless, there was a tendency to slightly or highly suppress biofilm formation. Interestingly, the inhibition of biofilm formation by ascorbic acid has been described before in biofilms of *Bacillus subtilis*, *Escherichia coli*, *Pseudomonas aeruginosa* (*Pandit et al., 2017*) as well as of methicillin-resistant *Staphylococcus aureus*

(*Mirani et al., 2018*). It should be noted that at higher concentrations, L-ascorbic acid has been reported as a possible adjuvant for antibiotic treatment of BV, playing a role in maintaining a low vaginal pH, which favors the recolonization of the vaginal environment with lactic acid-producing bacteria, decreasing, thereby, the risk of BV recurrence (*Krasnopolsky et al., 2013*; *Polatti et al., 2006*). Our data further expand these previous findings by demonstrating that, while sometimes favoring planktonic growth, L-ascorbic acid often impairs biofilm formation.

A limitation of this study was the fact that we only tested a yet unidentified *Gardnerella* sp. isolate, but at least three new species have been recently reported. Previously, we assessed biofilm formation by seven clinical isolates from BV-women and seven from healthy microbiota and found no significant differences between the ability to form biofilms by the 2 groups, using different growth media (*Castro et al., 2015*). We now know that from those 14 isolates, some belong to *G. vaginalis*, *G. leopoldii*, *G. piotii* and *G. swidsinskii* (*Castro et al., 2020*). As such, we hypothesized that the four *Gardnerella* species would have similar biofilm formation abilities in our growth medium of choice: NYC. To test this hypothesis, we selected one isolate of each species, previously found to form similar biofilms in sBHI (*Castro et al., 2020*; *Vaneechoutte et al., 2019*), and compared its biofilms with NYC medium. As shown in Fig. S2, all the tested species had a similar biofilm-formation ability as compared to *Gardnerella* sp. UM241, with *G. leopoldii* showing a slight decrease in biomass, but well within the expected variation found in different *Gardnerella* strains (*Castro et al., 2015*).

## CONCLUSIONS

Overall, our work has shed new light on the optimal conditions required for in vitro growth and biofilm formation of bacteria associated with BV. Although we tested nine different growth conditions, including a medium simulating genital tract secretions (mGTS), none of them is able to account for all growth factors present in the vaginal environment, including components of the host immune system, that are known to interfere in bacterial growth (*Castro, Jefferson & Cerca, 2018*). Nevertheless, this work highlighted that under the appropriate in vitro conditions, some of the most common species found in BV can form single-species biofilms, contrary to what was shown before (*Castro et al., 2020*; *Patterson et al., 2010*). NYC medium revealed to be an ideal candidate for future studies addressing multi-species biofilm formation since this growth medium allowed significant levels of single-species biofilm formation. Understanding microbial interactions that occur during BV development is crucial for the development of novel antimicrobial strategies, and future work will help to clarify some of these crucial interactions in multi-species biofilms.

### Funding

This work was supported by the research project [PTDC/BIA-MIC/28271/2017] under the scope of COMPETE 2020 [POCI-01-0145-FEDER-028271], supported by the Portuguese

Foundation for Science and Technology (FCT), and by the strategic funding of unit [UIDB/04469/2020]. Aliona S. Rosca received financial support from individual Grant [PD/BD/128037/2016]. Nuno Cerca received support from the National Institute of Allergy and Infectious Diseases (R01AI146065-01A1, granted to Christina A. Muzny, MD, MSPH). The funders had no role in study design, data collection and analysis, decision to publish, or preparation of the manuscript.

### Grant Disclosures

The following grant information was disclosed by the authors:
Portuguese Foundation for Science and Technology (FCT): PTDC/BIA-MIC/28271/2017, POCI-01-0145-FEDER-028271 and UIDB/04469/2020.
Individual Grant: PD/BD/128037/2016.
National Institute of Allergy and Infectious Diseases: R01AI146065-01A1.

### Competing Interests

The authors declare that they have no competing interests.

### Author Contributions

- Aliona S. Rosca performed the experiments, analyzed the data, prepared figures and/or tables, authored or reviewed drafts of the paper, and approved the final draft.
- Joana Castro performed the experiments, analyzed the data, authored or reviewed drafts of the paper, and approved the final draft.
- Nuno Cerca conceived and designed the experiments, authored or reviewed drafts of the paper, and approved the final draft.

### Data Availability

The raw data measurements related to the data are available in the Supplemental Files.

### Supplemental Information

Supplemental information for this article can be found online at http://dx.doi.org/10.7717/peerj.9917#supplemental-information.

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

# PeerJ

**Swidsinski A, Loening-Baucke V, Mendling W, Dörffel Y, Schilling J, Halwani Z, Jiang XF, Verstraelen H, Swidsinski S. 2014.** Infection through structured polymicrobial *Gardnerella* biofilms (StPM-GB). *Histology & Histopathology* **29**:567–587 DOI 10.14670/HH-29.10.567.

**Swidsinski A, Mendling W, Loening-Baucke V, Ladhoff A, Swidsinski S, Hale LP, Lochs H. 2005.** Adherent biofilms in bacterial vaginosis. *Obstetrics & Gynecology* **106(5 Pt. 1)**:1013–1023 DOI 10.1097/01.AOG.0000183594.45524.d2.

**Turovskiy Y, Cheryian T, Algburi A, Wirawan RE, Takhistov P, Sinko PJ, Chikindas ML. 2012.** Susceptibility of *Gardnerella vaginalis* biofilms to natural antimicrobials subtilosin, ε-poly-L-lysine, and lauramide arginine ethyl ester. *Infectious Diseases in Obstetrics and Gynecology* **2012(2)**:1–9 DOI 10.1155/2012/284762.

**Vaneechoutte M. 2017.** *Lactobacillus iners*, the unusual suspect. *Research in Microbiology* **168(9–10)**:826–836 DOI 10.1016/j.resmic.2017.09.003.

**Vaneechoutte M, Guschin A, Van Simaey L, Gansemans Y, Van Nieuwerburgh F, Cools P. 2019.** Emended description of *Gardnerella vaginalis* and description of *Gardnerella leopoldii* sp. nov., *Gardnerella piotii* sp. nov. and *Gardnerella swidsinskii* sp. nov., with delineation of 13 genomic species within the genus *Gardnerella*. *International Journal of Systematic and Evolutionary Microbiology* **69**:679–687 DOI 10.1099/ijsem.0.003200.

**Weeks RM, Moretti A, Song S, Uhrich KE, Karlyshev AV, Chikindas ML. 2019.** Cationic amphiphiles against *Gardnerella vaginalis* resistant strains and bacterial vaginosis-associated pathogens. *Pathogens and Disease* **77(8)**:ftz059 DOI 10.1093/femspd/ftz059.

**Wijesinghe G, Dilhari A, Gayani B, Kottegoda N, Samaranayake L, Weerasekera M. 2019.** Influence of laboratory culture media on in vitro growth, adhesion, and biofilm formation of *Pseudomonas aeruginosa* and *Staphylococcus aureus*. *Medical Principles and Practice* **28(1)**:28–35 DOI 10.1159/000494757.