# Peer review of "Evaluation of different culture media to support in vitro growth and biofilm formation of bacterial vaginosis-associated anaerobes"

_PeerJ, doi:10.7717/peerj.9917_

## Round 0.1 · original submission · Major Revisions

Dear Dr. Rosca and colleagues:

Thanks for submitting your manuscript to PeerJ. I have now received three independent reviews of your work, and as you will see, the reviewers raised some concerns about the research. Despite this, these reviewers are optimistic about your work and the potential impact it will lend to research on bacterial vaginosis. Thus, I encourage you to revise your manuscript, accordingly, taking into account all of the concerns raised by the reviewers.

Please improve the content and clarity of the figures and tables (see suggestions by the reviewers). Please also ensure that all appropriate references are included.

Please clarify the selection criteria for the anaerobes. A concern over using only PubMed as the only search engine is raised and potentially excludes other BV-associated microbes. Please quantify the frequency level (stating “very frequent” or “frequent” is misleading as there other BV microbes more prevalent in certain populations). Please specifically justify the inclusion of Lactobacillus iners.

While the concerns of the reviewers are relatively minor, this is a major revision to ensure that the original reviewers have a chance to evaluate your responses to their concerns. There are many suggestions, which I am sure will greatly improve your manuscript once addressed.

I look forward to seeing your revision, and thanks again for submitting your work to PeerJ.

Good luck with your revision,

-joe

Reviewer 1 ·

Basic reporting

The authors of the manuscript have long been working in the field of bacterial biofilms and reveal their significance in developing vaginal dysbiosis using in vitro models. In this manuscript, the authors test various culture media to analyze the ability of vaginal bacteria to grow in vitro. The selection of appropriate medium/a to generate suitable models would allow measuring the impact of other vaginal bacterial species earlier not tested. Evaluation of medium/a suitable for a generation multi-species biofilms allows modeling the strategies for the management and eradication of biofilms.
The aim of the manuscript is clear and unambiguous; the background has written very clear, and with deep knowledge in the field. The structure of the article is clear with sufficient tables and figures provided; raw data are included.

Experimental design

The experimental methods are well described, data from the technical replicates and independent assays are generated.

Validity of the findings

The data provided are well-grounded and statistically relevant. Conclusions are well-stated.

There is an issue to be addressed:
Table 1, footnote 2: for clarity, please, explain how do you know that strain UM241 “belongs to the cluster containing genomic species 1 to 4 as depicted in Figure 1 of Vaneechoutte et al., 2019, hence, likely genomic species 2 or 3”. Could you please provide any evidence for this statement in the article? UM241 strain does not belong to G. vaginalis, G. piotii, G. leopoldii, and G. swidsinskii as stated in Castro et al., 2020.

My suggestion:
Table 2: it is not clear what is the difference between medium sBHI and sBHI.Aa, BHV/ BHV.Aa etc. Could you please provide an explanation of Aa in e.g. a footnote? In the Methods section (subsection Bacterial species and growth conditions, line 105) designation of Aa is lost among the recipes of culture media.

Additional comments

The article is well written and it is of high importance for the generation of in vitro models analyzing microbial species interaction in vaginal.
There is an issue to be addressed:
Table 1, footnote 2: for clarity, please, explain how do you know that strain UM241 “belongs to the cluster containing genomic species 1 to 4 as depicted in Figure 1 of Vaneechoutte et al., 2019, hence, likely genomic species 2 or 3”. Could you please provide any evidence for this statement in the article? UM241 strain does not belong to G. vaginalis, G. piotii, G. leopoldii, and G. swidsinskii as stated in Castro et al., 2020.
My suggestion:
Table 2: it is not clear what is the difference between medium sBHI and sBHI.Aa, BHV/ BHV.Aa etc. Could you please provide an explanation of Aa in e.g. a footnote? In the Methods section (subsection Bacterial species and growth conditions, line 105) designation of Aa is lost among the recipes of culture media.

·

Basic reporting

Evaluation of different culture media to support in vitro growth and biofilm formation of bacterial vaginosis-associated anaerobes

The author(s) tried to grow BV-associated anaerobes in different culture media. However, the selection criteria for the anaerobes was not clear. Using only PubMed as the only search engine will exclude other BV-associated microbes. The frequency level was not quantified. Stating very frequent or frequent is misleading as there other BV microbes more prevalent in certain populations.

I wonder why the author(s) did not reference the original source for the
Chemically defined medium simulating genital tract secretions (Stingley et al., 2014). Stingley et al are not the original developers of the medium but may have modified it by adding proteose peptone and ascorbic acid which have been documented by several authors.
These authors should be the right reference.
Geshnizgani AM, Onderdonk AB. Defined medium simulating genital tract secretions for growth of vaginal microflora. J Clin Microbiol 1992;13236

Experimental design

The inclusion of Lactobacillus iners as a BV-associated microbes is still under debate as the microbes is very much found in all healthy women without symptoms.

The experimental design should have included vaginal secretion from BV subjects to see whether these microbes could be isolated from the various media used in this experimental study.

On the biofilm assay, the author(s) should explain why they did not use confocal microscopy to visualize the biofilm formed. Using the crystal violet has low sensitivity.

Validity of the findings

This statement "A particularity of NYC medium, compared to the other tested media, is the presence of proteose peptone no. 3, which has been described by the manufacturer as offering high nutritional benefits to fastidious anaerobic species by providing the necessary amount of nitrogen, carbon, amino acids, and essential growth factors" is also misleading as this factor was not determined.

This may not the only reason as proteose peptone is present in other Vaginally defined medium 0.5% proteose peptone (VDMP).

Additional comments

In real clinical setting, how can the author(s) defend this statement?
"NYC medium revealed to be an ideal candidate for future studies addressing multi-species biofilm formation since this growth medium allowed significant levels of single-species biofilm formation" In reality, BV forms polymicrobial biofilms. There is no single-species associated with BV condition.

Reviewer 3 ·

Basic reporting

1. The authors reference Diop et al., 2019 (line 85) and list the bacteria in Table 1 as the rationale for selecting bacteria tested in this manuscript. Additional reasoning on why these bacteria were selected is needed. For example, why was Lactobacillus iners selected? Most readers outside the immediate field consider lactobacilli as being associated with health. It would be important to highlight that L. iners is present in high relative abundance and concentrations in women with and without BV and include appropriate references that discuss this point specifically (Petrova et al. 2017; Srinivasan et al., 2012). Likewise, given that Gardnerella vaginalis have now been classified as at least four different species, why was only one Gardnerella species selected for these studies? There is limited information as to which Gardnerella species may be important in BV, and hence it would be important to present data for all four Gardnerella species. It is also unclear what search strings were used in Pub Med to result in the frequency of articles that were listed in Table 1 and they need to be clarified.

7. There are areas where language can be clarified, improved or made concise. Examples below.
- Line 119-120: Combine both sentences to make concise
- Line 147: No need to emphasize that “our results demonstrated” that the different BV anaerobes had different nutritional requirements for optimal growth – this is to be expected and can be stated as such.
- Line 168: similar, not similarly
- Line 201: rephrase

Experimental design

2. Growth was assessed using optical density measurements and reported as a ratio of OD of the culture media relative to values at time 0. In many instances, the value was close to 1 (0.8 to 1.5). This would represent an OD of 0.1 at time 0 and 0.1 after the incubation period (can go up to 0.15) which really does not represent any growth. For example, an OD ratio of 1.4 (change in OD from 0.1 to 0.14) for Mobiluncus curtisii in NYC does not seem like growth. Measuring CFU/mL using colony counts would have been a better approach to evaluate growth and viability. I would recommend showing these data for at least the medium that the authors are proposing as optimal for growth (NYC).
3. The authors nicely show that ascorbic acid may inhibit growth as is the case with Prevotella bivia in SB and SB.Aa medium. Practically no growth was observed for any bacterium in mGTS medium. Did the authors consider preparing mGTS medium without ascorbic acid to determine if the primary reason for lack of growth was the presence of ascorbic acid in mGTS medium?
4. Table 3 shows growth and p-values represented in the form of letters to show comparisons between different referent groups. It would be easier if only one referent group was shown in this table and another table showed the different comparisons as a heatmap with actual p-values.
5. Lines 241-245: The authors propose that ascorbic acid may play a role in recolonization of the vaginal environment with lactic acid-producing bacteria. Did the authors consider testing the growth of Lactobacillus crispatus or L. jensenii (lactic acid producers) in media with ascorbic acid to test this hypothesis?
6. How many times were bacterial cells passaged before a new stock was obtained from the freezer?

Validity of the findings

No comment

Additional comments

The manuscript by Rosca et al. evaluates 5 different culture media with and without supplementation with ascorbic acid for growth of 6 bacterial species that have been associated with the common dysbiotic condition, bacterial vaginosis. One goal was to identify a cultivation medium that can support in vitro biofilm formation for all bacterial strains tested such that future co-culture experiments evaluating multi-species interactions can use a single cultivation medium in experiments. While the data presented will be of value for investigators interested in conducting in vitro mechanistic studies of BV-associated anaerobes, I have some questions regarding the study which are listed in the sections above.

---

## Round 0.2 · accepted · Accept

Dear Dr. Rosca and colleagues:

Thanks for revising your manuscript based on the concerns raised by the reviewers. I now believe that your manuscript is suitable for publication. Congratulations! I look forward to seeing this work in print, and I anticipate it being an important resource for groups studying bacterial vaginosis. Thanks again for choosing PeerJ to publish such important work.

Best,

-joe

Reviewer 1 ·

Basic reporting

No comment

Experimental design

No comment

Validity of the findings

No comment

Additional comments

I have read the rebuttal letter and the revised manuscript. All my comments are addressed in the revised manuscript.

·

Basic reporting

No comment

Experimental design

The authors have made and supported their case

Validity of the findings

The authors have provided extra points including references to prove their findings.

Additional comments

In my opinion, authors did go extra mile to provide answers to the queries. Good job!!!

Reviewer 3 ·

Basic reporting

The edits made to improve clarity is appreciated.

One quibble is on line 90. L. iners does not necessarily play a controversial role, just we have a poor understanding of the role of L. iners in bacterial vaginosis as it is present both in women with and without BV.

Perhaps re-state along the lines of: L. iners was also included in the study as this bacterium is present in women with and without BV and its role in the vaginal microenvironment is poorly understood.

Also Line 246: I believe the word here should be "recipe" not "receipt."

Experimental design

I applaud the authors on being responsive to the comments made by the authors including performing additional experiments

1. to evaluate whether there were any differences in biofilm mass due to specialized components in NYC with proteose peptone no. 3 compared with NYC with peptone from meat

2. to evaluate differences if any in biofilm formation by the validly named Gardnerella species including G. piotii, G. leopoldii, G. vaginalis, and G. swidsinkii

Validity of the findings

No additional comments